# Graphene and Natural Products: A Review of Antioxidant Properties in Graphene Oxide Reduction

**DOI:** 10.3390/ijms25105182

**Published:** 2024-05-09

**Authors:** Filipe Kayodè Felisberto dos Santos, Antônio Augusto Martins Pereira Júnior, Arquimedes Lopes Nunes Filho, Clícia Joanna Neves Fonseca, Daysianne Kessy Mendes Isidorio, Filipe de Almeida Araújo, Pablo Henrique Ataide Oliveira, Valdir Florêncio da Veiga Júnior

**Affiliations:** 1Chemical Engineering Section, Military Institute of Engineering, Rio de Janeiro 22.290-270, Brazil; 2Higher School of Engineering, Technology and Innovation, University of the Federal District Professor Maia Nunes, Brasília 70.635-815, Brazil; antonio.pereira@undf.edu.br; 3Postgraduate Program in Materials Science and Engineering, Military Institute of Engineering, Rio de Janeiro 22.290-270, Brazil; arquimedesfilho94@ime.eb.br (A.L.N.F.); clicia.fonseca@ime.eb.br (C.J.N.F.); 4Department of Metallurgical and Materials Engineering, Federal University of Rio de Janeiro, Rio de Janeiro 21.941-901, Brazil; daysiannekessy@metalmat.ufrj.br; 5Postgraduate Program in Materials Science and Engineering, Federal University of São Carlos, São Carlos 13.565-905, Brazil; filipearaujo@estudante.ufscar.br; 6Higher Education Department of Education, Federal Institute of the North of Minas Gerais, Bom Jardim 39.480-000, Brazil; pablo.oliveira@ifnmg.edu.br

**Keywords:** natural products, graphene, compounds, antioxidants, health

## Abstract

This review article addresses the antioxidant properties of different natural products, including ascorbic acid, gallic acid, oxalic acid, L-glutathione (GSH), bacteriorhodopsin, green tea polyphenols, glucose, hydroxycinnamic acid, ethanoic acid, betanin, and L-glutathione, in the reduction of graphene oxide (rGO). rGO can cause damage to cells, including oxidative stress and inflammation, limiting its application in different sectors that use graphene, such as technologies used in medicine and dentistry. The natural substances reviewed have properties that help reduce this damage, neutralizing free radicals and maintaining cellular integrity. This survey demonstrates that the combination of these antioxidant compounds can be an effective strategy to minimize the harmful effects of rGO and promote cellular health.

## 1. Introduction

The advancement of studies related to nanotechnology has developed products and by-products with a multitude of technological applications, among which, graphene stands out. Graphene consists of a two-dimensional (2D) network of carbon atoms (C) arranged in densely packed hexagonal structures, whose thickness is of the order of 0.15 nm (diameter of the carbon atom) and density of 0.3 mg/cm. Having said this, it is noted that graphene is a thin, light, flexible, and highly resistant nanomaterial [1,2].

Regarding its properties, excellent electrical, mechanical, and thermal properties can be highlighted, resulting, above all, from its nanostructural and electronic characteristics. For example, it has a high Young’s modulus (1.0 TPa), higher than that of many steels, indicating that high mechanical stress is necessary to cause deformation [3]. Furthermore, it presents an extraordinary mobility of electrons due to the rapid transport of charges between the layers and the large surface area, which results in high electrical conductivity at room temperature, making graphene very suitable for use in supercapacitors [4]. Furthermore, it also has excellent thermal conductivity and can be transparent, making it suitable for designing solar panels and touch screens. The small areas of each carbon hexagon are responsible for the high impermeability of graphene, which can be used as a small network capable of holding gases that easily leak from their containers, such as hydrogen gas. In addition to being extremely resistant, graphene is very light: its density is 0.77 g/m^2^, that is, around a thousand times lighter than a sheet of paper [4]. These well-known characteristics have encouraged studies and research for into graphene’s application in energy storage and conversion and in environmental applications, as well as in the assessment of toxicity in biological systems [5,6,7].

In electronics, graphene’s high electrical conductivity and transparency have led to advancements in flexible and transparent electronic devices. Moreover, in energy storage, graphene-based materials show potential for enhancing the performance of supercapacitors and batteries due to their large surface area and excellent conductivity [8,9]. In the realm of medical science, the exploration of nanomaterials (such as graphene and its derivatives) has emerged as a groundbreaking avenue in disease management, offering a promising application in healthcare. Taking advantage of graphene’s unique characteristics, researchers are developing advanced drug delivery systems capable of precisely targeting diseased cells while minimizing collateral damage to healthy tissues [10]. Furthermore, graphene-based biosensors exhibit exceptional sensitivity, enabling the early detection and monitoring of various diseases, from cancer to neurological disorders [11,12]. Additionally, graphene’s antimicrobial properties hold potential in combating antibiotic-resistant pathogens, presenting a promising solution to the global healthcare challenge of infectious diseases [13].

The introduction of reduced graphene oxide (rGO) into cells can induce oxidative stress. In the context of oxidative stress-related biological outcomes, two primary mechanisms dominate. The first mechanism centers on the production of reactive oxygen species (ROS) and reactive nitrogen species (RNS), leading to biomolecular oxidation, consequentially resulting in cellular dysfunction and mortality. Simultaneously, the second oxidative stress mechanism revolves around the perturbation of redox signaling pathways. Consequently, antioxidants play a crucial role, either by scavenging ROS and RNS [14,15] or by modulating redox signaling pathways [16].

On the other hand, the method of obtaining rGO through different synthesis can be worrying due to the risk of toxicity, such as reduction with hydrazine [17]. Another point to be highlighted is that GO has been shown to be a biocompatible material; however, when inserted into cell culture medium, there is a loss of cell functionality in 20% and 50% when 20 mg.mL^−1^ and 50 mg.mg^−1^, respectively, is added [18]. Furthermore, studies report that graphene-based particles present in the blood can travel to different organs in our body and can reach the nervous system, and when ingested intravenously or orally, they mainly reach the kidneys, lungs, and liver, leading to inflammation [19,20]. Another publication with in vivo studies shows that prolonged exposure to GO in doses of 25 micrograms L^−1^ caused opacity in the animals’ eyeballs, in contrast to RGO, which exhibited biosafety [21]. 

Concern about environmental issues, especially in terms of minimizing or eliminating harmful chemical substances or those that result in toxic waste, has encouraged many researchers to obtain reduced graphene oxide, and this method has been called green reduction of graphene oxide. Among the various approaches, one can include the use of natural reducing agents, such as plant extracts, and reduction methods using water as a solvent. However, the big challenge of this green route is related to efficiency when compared to the synthesis route with hydrazine, for example, and obtaining an rGO with good properties. Among all the green routes mentioned by [22] the one where ascorbic acid was used to obtain RGO showed the most promise. In addition to being a recent process, it has shown excellent results compared to hydrazine methods; however, more research is needed in order to model and find a better optimization process for this route with changing parameters and conditions in search of large-scale production [22,23,24].

Given the relevance of graphene to industry and its promising properties, synthesis methods for this material are the target of research, as the large surface area and oxygen-containing functional groups of graphene oxide (GO), such as phenol, hydroxyls, epoxy and carboxylic groups, make it ideal for providing high substance loading efficiency, good dispersion, and easy functionalization [25]. Among those already used are the micromechanical exfoliation of highly oriented pyrolytic graphite (HOPG), chemical vapor deposition (CVD), and epitaxy. These techniques produce different qualities of graphene with relative structural perfection and excellent properties. A highly relevant process for obtaining graphene consists in reducing its oxide, as it is the most reliable and commercially viable method among those currently known for this purpose [26]. Reduction can have different approaches, such as thermal, photochemical, biological, and chemical. In the latter, the substance interacts with the oxygenated groups present in GO, releasing water, which will later be used in a heat treatment to form the π bonds of the graphene structure [27].

In this context, different natural products emerge as potential agents for reducing GO, such as antioxidant compounds and their great capacity for removing and/or stabilizing oxygenated groups present in GO. Substances such as ascorbic acid, gallic acid, oxalic acid, and others have been presented as important natural sources for obtaining graphene [28,29,30]. This review article presents a detailed and comprehensive analysis on the potential applications of natural antioxidants in the graphene oxide reduction process. This review stands out for its depth and scientific rigor, offering a solid foundation of knowledge that not only expands the understanding of reduced graphene oxide, but also sheds light on the effective use of natural reducing agents. Furthermore, this study also investigates the environmental and biological implications of these interactions, suggesting promising paths for future research and technological applications that combine sustainability and innovation in materials engineering.

### 1.1. Chemical Reduction

The formation of GO involves the reaction of graphite with strong oxidants such as sulfuric acid, nitric acid, potassium chlorate, and potassium permanganate. The introduction of functional groups containing oxygen, through complete oxidation, results in an increase in the spacing between the graphite layers, subsequently making it possible to obtain graphene [30]. SiO_2_ surfaces at 150 °C are spray-coated and the graphite oxide can be chemically reduced using hydrazine. However, this reducing agent is toxic and dangerous, and its use must be carried out with extreme care and in minimum quantities. On the other hand, natural reducing agents can act in a safer and more environmentally friendly way with regard to graphene production, thus enabling the development and compatibility of more sustainable and less expensive synthesis routes (Figure 1) [31,32].

### 1.2. Graphene Oxide-Reducing Agents from Natural Products

Graphene is a two-dimensional (2D) material composed of a single allotropic carbon sheet with sp2 hybridization. Producing graphene in large quantities is one of the biggest challenges in the materials sector. Of the various methods that have been explored, GO reduction is considered the most reliable and commercially viable method [30]. GO can exhibit exceptional electronic, mechanical, and thermal properties similar to graphene, reducing treatments that transform GO into reduced graphene oxide (RGO) [33]. Various reduction methods have been developed, such as chemical, biological, photo-mediated, and thermal reduction [30]. Some substances commonly used for these reductions are hydrazine, hydroquinone, and sodium borohydride. Unfortunately, these are dangerous chemicals in terms of human health and environmental awareness. Thus, implementing sustainable alternatives to these reducing agents would be highly beneficial, in turn facilitating the large-scale production of GO graphene for multiple applications [31]. The use of ascorbic acid as a GO reducing agent has been explored for various purposes and using the most varied combinations of analysis techniques, as can be seen in Table 1.

Next, in Table 2, oxalic acid is presented, a common reducing agent due to its greater reducing power when compared to other acids such as acetic acid.

Gallic acid (Table 3) and GSH (Table 4) are reducing agents that have been recently exploited mainly as precursors for nanoparticle deposition, sensory applications, and electrodes.

Polyphenols (Table 5) and glucose (Table 6) are natural reducers whose use has grown in recent years. This is due to their characteristics and applications, such as drug delivery systems and treatments for specific conditions of the human body.

#### 1.2.1. Ascorbic Acid

Ascorbic acid (AA), widely known as “vitamin C” (Figure 2), is a ketolactone found mainly in citrus fruits, such as oranges and lemons, but is also available in other natural sources, such as acerola, kale, tomatoes, broccoli, strawberries, and peppers [48]. When applied as an antioxidant, its main function is to remove oxygen, capturing it in the reaction medium, making it unavailable to propagate auto-oxidation [49]. The removal of oxygen functional groups by AA is relatively easy, as a small amount of AA could carry out the reduction in a short time, resulting in a product with a high carbon/oxygen ratio. It can also present another activity, in which the molecule donates electrons to radicals and which will stabilize through resonance, preventing such radicals from reacting with oxygen or other molecules to form more reactive radicals [50].

Its antioxidant properties as a scavenger of free radicals are related to its ability to form a stabilized radical, allowing the formed ascorbate to react with more reactive molecules. As such, the ascorbate both suppresses free radicals and stimulates the oxidant system on the cells while acting on antioxidant enzymes as a ROS eliminator [51].

Furthermore, its lower toxicity and greater biocompatibility compared to other common reductants such as hydrazine or sodium borohydride make ascorbic acid safer and more sustainable for applications in biotechnology and other areas. Its wide availability, stability, and antioxidant properties are also recommended for its effectiveness and practicality, protecting the formed graphene from oxidative manipulation. Therefore, ascorbic acid proves to be an efficient reducing agent not only due to its physicochemical characteristics, but also due to its environmental and safety advantages, making it an ideal choice for the sustainable production of graphene in scientific and industrial applications [52].

#### 1.2.2. Gallic Acid

Gallic acid (3,4,5-trihydroxybenzoic acid) (GA) (Figure 3) is a polyphenol, naturally found in teas, grapes, raspberries, blackberries, blueberries, and other fruits, in addition to being present in honey and wine. Furthermore, it is present in some plants, such as oak (*Quercus robur*) and Portuguese chestnut, as this species is known in Brazil (*Castanea sativa* L.) [53]. The substance is characterized as being a primary antioxidant, that is, it promotes the removal or inactivation of free radicals present in the reaction medium through the donation of hydrogen [54]. When used to reduce GO, gallic acid proves to be a compound with promising results in the process. The phenolic hydroxyl in its structure reacts with the epoxy group present in GO, to form a hydroxyl. The carbon of this newly formed hydroxyl reacts with another hydroxyl, creating an intermediate product and water. This intermediate product is subsequently subjected to elimination reactions to form the sp2 hybridized structure [55]. 

Faced with the need to develop a low-cost, high-efficiency, and environmentally friendly method that is capable of balancing graphene quality and production efficiency, several methodologies were created that apply GA to reduce GO, such as the GOAG nanocomposite, an anticancer nano dose delivery system using GO as a nanocarrier for an active anticancer agent, AG. When GOAG was applied to liver cancer cells, it showed an inhibitory effect on the growth of cancer cells without affecting normal cell growth [56]. Another approach to chemical reduction of GO using GA is the production of hybrid papers, multicarboxylic carbon-walled nanotubes (AG-rGO/MWCNTs), a mechanically robust material with gravimetric capacitance and ultra-high volumetric capacitance, which has cyclic stability capacity for capacitive electrochemical energy storage. Therefore, high-performance electrodes can be constructed using AG-rGO combined with carbon nanotubes. The results of these research studies are highly encouraging to continue exploring the potential of GO reduction by AG as well as other environmentally friendly and applicable reducers [57].

#### 1.2.3. Oxalic Acid

Oxalic acid (OA) or ethanedioic acid (Figure 4) is an organic acid with a simple structure. Members of the plant kingdom, including land plants and woody trees such as conifers, accumulate and retain oxalic acid in their leaves, needles, and twigs. OA stands out for being present in broccoli, cabbage, and cauliflower leaves when subjected to cooking, as well as in the outer layers of cereal grains, such as wheat bran [58]. OA stands out for preserving quality and postponing the physiological effects of fruit aging. Although its mechanism of action has not yet been elucidated, the substance is related to preventing the accumulation of reactive oxygen species in the reaction medium [59].

Furthermore, its action in reducing GO occurs by removing oxygenated groups from GO, releasing H_2_O. A hydroxyl group is protonated in a basic sodium oxalate medium, producing water and forming the carbocation. The oxalate ions then interact with the neighboring hydrogens to form the π bonds of the graphene structure [59]. Based on previous studies, rGO reduced using oxalic acid (rGO/AO) has a rough surface, folded edges, ripples, a multilayer structure, tangles of graphene layers, and the presence of ultra-thin and transparent graphene sheets as its characteristic morphological profile. Furthermore, in view of this, it is noteworthy that the folds in the edge regions confer high tenacity to the material, therefore leading to excellent mechanical properties [60].

rGO/OA has a structure based on flat layers with agglomerated graphene sheets, whose dimensions vary from ≅8 to 10 μm. However, there are reports of the synthesis of large-sized GO sheets reduced with OA in the range of 0.5 to 10 μm. rGO/AO, in general, has a porous nature with pore sizes of the order of ≅2–5 μm. The process of reducing GO with OA leads to a rapid removal of oxygen atoms from the graphene basal plane, thus allowing the modulation of the desired morphology (dendrites, nanosheets, and nanorods) [60]. In addition, it is worth highlighting that OA (an oxidized form of ascorbic acid) confers greater stability to rGO. This fact arises from the affinity of this compound to bind hydrogen atoms from residual oxygenated functional groups, thus interrupting π–π between the rGO sheets and the formation of agglomerations [61]. It is also noteworthy that graphene obtained via chemical exfoliation with OA can present a nanostructural configuration based on monolayers or even multiple overlapping carbon layers [60,61]. 

#### 1.2.4. L-Glutathione (GSH)

L-glutathione (Figure 5), commonly known as GSH, is the main thiol-containing tripeptide of low molecular mass (307.32 g mol^−1^), composed of glutamate, cysteine, and glycine [62]. It is found in most living cells of bacteria, mammals, plants, and fungi. GSH is often used for its antioxidant and anti-inflammatory properties and is available in supplement form but can also be obtained from dietary sources. Foods rich in GSH include Brazil nuts, broccoli, Brussels sprouts, kohlrabi, cauliflower, cabbage, onions, and some mushrooms [63]. Among its many functions, it can also protect important cellular compounds from oxidative stress caused by free radicals, air pollution, heavy metals, and the harmful effects of radiation, by acting as a scavenger, neutralizing reactive oxygen species (ROS) [62]. Thus, by incorporating GSH into the preparation of GO, it simultaneously reduces and stabilizes the monolayers of GO nanosheets [64]. This occurs because the electrostatic repulsion between the terminal carboxylic acid of GSH and GO provides sufficient negative charge, promoting stable dispersion in aqueous solutions and polar aprotic solvents, thus avoiding agglomeration and precipitation [63].

L-glutathione (GSH) is a natural antioxidant that allows the reduction of multiple types of oxygenated compounds [63]. Thus, the oxygenated form of L-glutathione has external carboxylic groups with a negative charge, which generates electrostatic repulsion, promoting a stable dispersion of graphene, inhibiting the regrouping of its nanosheets [64]. In recent works [65], the chemical reduction of GO was carried out using GSH. By observing the FTIR spectrum, this reduction can be checked by analyzing the absorption peaks (Figure 6). The peaks corresponding to the oxygenated functional groups were reduced, with the functional groups disappearing, leaving only one at 1534 cm^−1^, which relates to the graphitic structure of graphene nanosheets [66].

Looking at Figure 7, the morphology of GO can be seen, characterized by thin sheets of graphene, whose appearance is characterized by wrinkles and folds. Furthermore, it appears that the material has a multilayer configuration, that is, densely packed areas of stacks and agglomerations.

#### 1.2.5. Bacteriorhodopsin (bR)

Bacteriorhodopsin (bR) is a protein membrane found in salt marshes of the Archaea Halobacterium salinarum, and consists of photoactive fractions called retinal chromophores (C_20_H_28_O) in the middle of a channel formed by seven transmembrane alpha-helices (A–G) [67]. As it is a proton pump powered by light, when light of one wavelength is irradiated, it can convert the bound electrons into free radicals [68]. Thus, its use as a GO reduction mechanism involves the distribution of bR protons between the GO sheets. When yellow light (≅80 mW cm^−2^, 530 nm) is irradiated, the oxygen group interacts, forming water molecules and restoring the sp2 hybridized carbon structure, therefore resulting in rGO [69].

Despite the sustainable potential of bR as an effective graphene reducer, there are few studies in the literature that report its morphology. Only one study was found on the morphology of bR/rGO. Researchers evaluated bR molecules as light-driven proton pumps for the reduction of single layers of GO. In this sense, through AFM, the topography and distribution of the height profile of the graphene and bR sheets can be observed. Furthermore, there was an absence of direct contact between bR molecules and GO sheets [70]. 

#### 1.2.6. Green Tea Polyphenols

Green tea polyphenols (GTPs) are a group of compounds that increase human metabolism, resulting in many health benefits. GTPs include several phytochemicals such as epicatechin (EC), epicatechin gallate (ECG), epigallocatechin (EGC), and epigallocatechin gallate (EGCG) (Figure 8). Among these, EGCG has the highest concentration of catechins, indicating its high antioxidant functionality [71]. In its chemical structure, tea polyphenols (TPs) are able to donate an electron or hydrogen atom, making it an effective reducing agent.

Thus, the reaction process between the phenolic hydroxyl groups of Tps and the epoxy group in GO forms a hydroxyl group by opening the nucleophilic ring. These hydrogen ions act as catalysts to promote the dehydration of OG, resulting in the formation of water as a byproduct and the sp^2^ hybridized carbon structure of rGO (Figure 9).

GTPs, up to 30% of dry weight, have the highest concentration of all catechins in GT (≅9–13% of dry weight) and are known to be the most powerful active antioxidant components of GT as a reductant during the synthesis of rGO [72]. Thus, it is observed that the mechanism for eliminating oxygenated functional groups from GO via these polyphenols is reflected in the internal folds and pores of the rGO structure. Furthermore, it appears that this process favors the production of nanosheets with a large structure, providing a specific surface, which is important for fixing metals [73].

The removal of electrons from the catechin rings makes the hydroxyl bonds more acidic, so the hydroxyl bonds are organized to split two protons and function as a nucleophile. The GO leaves mainly contain two types of reactive oxygen species: epoxide and hydroxyl. Both epoxide and hydroxyl bonds can be opened by means of nucleophilic attack by the oxygen anions of catechins [74]. The reduction must also be followed by the dehydration process, carried out by absorbing an electron from the environment. Next, the catechins are oxidized into benzoquinone-type products. Benzoquinone-type products can be absorbed in leaves reduced by π–π interactions between the aromatic rings of the products and leaves [75].

The GO suspension is reduced in a few minutes and has excellent dispersibility and chemical stability for a period of six months. This stability can be attributed to the fixation of oxidized GTPs on the surface of the reduced leaves [76]. Temperature directly affects the thermal stability of rGO using green tea polyphenols, as there is a weight loss of approximately 22% in the temperature range between 150 and 230 °C. The significant weight loss at this temperature was associated with the decomposition of the oxygen functional group into GO, and there was additional weight loss between 450 and 565 °C, mainly due to the burning of the graphitic regions [76].

#### 1.2.7. Glucose

According to the literature, carbohydrates, also called glycans, are one of the most prevalent biopolymers in nature, and one of the three essential molecules of life alongside polynucleotides and polypeptides. The carbohydrate molecule’s presence is ubiquitous, found in all living beings, from animals and plants to microbes. Furthermore, its function is to play significant roles in physiological processes. In essence, the carbohydrate molecule is composed of carbon, hydrogen, and oxygen atoms, with the general chemical formula (CH_2_O)n. In this sense, monosaccharides such as glucose (GI) are important sugars for the energy metabolism of most living organisms. Furthermore, it is commercially available in the form of a white, crystallized, sweet, and odorless powder, with the formula C_6_H_12_O_6_ and a molecular weight of 180.16 g/mol [77].

With reducing capacity and non-toxic properties, GI works as a reducing agent, but with less efficiency when used to reduce GO. However, by adding elements such as ammonia, which raises the pH of the reaction or catalysts such as iron foil (Fe), it is possible to increase GI’s reducing functionality. In ammonia solution, GI is oxidized by GO and forms aldonic acid, which can be further converted into lactones. As a result, oxidized glucose products containing hydroxyl and carboxyl groups can form hydrogen bonds with residual oxygen functionalities on the GO surfaces to reduce to a stable glucose-rGO colloidal dispersion [78]. After the reduction of GO by glucose, the color of the suspension changes from light brown to black. This color change can be attributed to the partial restoration of the π–π structure between the leaves due to deoxygenation. This causes the composition of wrinkles at the edge of the rGO sheet, and further reduction can accommodate the properties of the wrinkles as well as their shrinkage [79].

The number of wrinkles present in GO nanoplates can contribute to the dissipation of frictional energy from the interface and improve the vibration damping performance. However, atomic force microscopy (AFM) shows that wrinkles with broad coefficients of friction (COFs) play an important role in delaying sliding events between rGO sheets [80]. Another morphological feature of glucose-reduced GO is the formation of a thick flake-like structure that can result in the agglomeration of much thinner flakes. This is due to end-to-end interactions between oxygen functional groups through hydrogen bonds. It is possible to visualize, in some regions, a small number of carbon spheres that contain a hydrophobic core and a hydrophilic shell [80].

The sugar-assisted thermal reduction of GO is applied to synthesize three-dimensional porous graphene (3D-PG), which can be used as an electrode for supercapacitors. It exhibits capacitance of about 115 F/g at a scan rate of 10 mV/s, and the addition of glucose in the GO reduction process strengthens the interconnectivities between the graphene sheets, making them stick together tightly and further accelerating cargo transport [81]. A green approach is applied to the fabrication of sterile nanometer-sized rGO (nrGO) for photothermal therapy and the delivery of drugs using pure glucose as a reducing agent. nGO reduced in pure glucose restores the conjugated structure and enhances the photothermal effect, making nGO more sensitive to the reduction carried out by glucose aldehyde groups. The oxidation of nitronium in OG promotes the reduction of nrGO to glucose and the degree of biocompatibility. Furthermore, the resulting nrGO has a 317% (*w*/*w*) loading of doxorubicin (DOX), a broad-spectrum chemotherapeutic agent known to adsorb on graphene-based materials via hydrophobic interactions and π–π stacking. And the release of DOX from the nrGO/DOX complex can be effectively enhanced by acidic conditions, GSH concentration, and heating, making nrGO a promising synergistic nano-platform for photothermal therapy and the controllable delivery of drugs, which can be applied to the field of nanomedicine [81].

Glucose added to the solution can increase the crosslinking of graphene sheets and the deoxygenation of GO sheets in the presence of metals such as Fe. This helps in recovering the graphitic structure of GO sheets during reduction [81]. The new findings provide relevant information on the green reduction of GO by sugarcane waste and its use to remediate metal ions [81,82]. These studies show the effectiveness of GO reduction by reducing sugars, such as glucose, as well as the applicability of rGO in environmental bioremediation. Sugarcane bagasse is mainly composed of hemicellulose (25–35%), lignin (15–35%), and cellulose (40–45%), and thermochemical treatment breaks the lignocellulosic bonds and partially solubilizes the polysaccharides to release sugars, such as glucose and sucrose [83]. Some applications require rGO with specific functional groups and this can be achieved by a simple one-step hydrothermal method to produce partially reduced GO rich in carbonyl functional groups using glucose as the sole precursor. The carbonyl functional groups present in the sample allow greater functionalization and provide good dispersibility in water, making them useful in biomedical and water treatment applications [83].

#### 1.2.8. Hydroxycinnamic Acid

The compound 3,4-hydroxycinnamic acid (C_9_H_8_O_4_), also known as caffeic acid (CA) (Figure 10) or 3,4-dihydroxycinnamic acid, consists of an aromatic organic compound with a mixed and unsaturated carbon chain consisting of cinnamic acid and hydroxylated benzene [84]. This biomolecule belongs to the group of polyphenols, and more precisely, to the chemical class of phenolic acids. Caffeic acid, as well as its derivatives, are biologically active substances of great importance regarding the tolerance and control of biotic and abiotic stresses in plants, such as, for example, pathogen attacks, low- and high-temperature stress, ultraviolet (UV) light stress, drought stress, heavy metal stress, and salinity stress. It is noteworthy that this phytonutrient is one of the products generated during the secondary metabolism of plant species (phenylpropanoid pathway) [85].

Furthermore, it is worth mentioning that this substance can be found in abundance in plants, fruits, and vegetables, of which the following stand out: olives, coffee, sage, oregano, mint, rosemary, thyme, sunflower seeds, potatoes, carrots, and burdock (sticky grass) [86]. Caffeic acid (CA) can also be isolated from wines, teas, propolis, and vegetable residues. CA and its derivatives (caffeoylquinic acids and dicaffeoylquinic acids), as well as chlorogeneic acid (an ester formed between caffeic acid and quinic acid), are responsible for the enzymatic browning of plant species, thus being natural antioxidant agents [87]. 

Regarding its use in the production of rGO, caffeic acid is usually used in hydrothermal reduction processes. When used as a reducing agent in such a process, it allows rGO with a morphology consisting of stacked agglomerates to be obtained [88]. Such morphological behavior comes from the hydrothermal process, as the removal of the oxygen group during reduction tends to provide a hydrophobic effect to the rGO layers, agglomerating to reduce free energy, due to hydrogen bonds between the layers. The reduction effect tends to affect the arrangement of the rGO layers, where the previously stacked layers become folded and curled after hydrothermal reduction, an effect corresponding to an attempt by the structure to become thermodynamically stable. The reduction effect tends to reduce the distance between rGO planes, a behavior caused by the reduction in the presence of oxygen in the structure. During reduction, variations in acid proportions tend to promote the formation of nanoparticles, with a reduction in particle size in accordance with the reduction in the proportion of caffeic acid. This behavior occurs due to the presence of rGO, which prevents particle growth [89].

Among the class of green reducing agents, caffeic acid has one of the greatest reducing effects, when used alone; at a C/O ratio of 7.15 during reduction, a low amount of rGO layers are produced, maintaining the standard structure of wrinkling [89].

#### 1.2.9. Ethanoic Acid

Ethanoic acid (CH₃COOH) or acetic acid (AA) is an organic compound with a saturated and straight chain, and it is an important chemical product widely used in the food industry [90]. Due to its characteristics and properties, this substance has traditionally been used in food preservation, as well as being a solvent or intermediate ingredient in a variety of chemical products of commercial interest, among which the following stand out: vinyl acetate monomers (VAM), cellulose acetate (anhydrous acetate), acetate ester, and terephthalic acid (TPA), which is the raw material of polyethylene terephthalate (PET) [91].

In general, AA is produced through an aerobic fermentation process, whose synthesis route is based on the oxidation of ethanol present in fruits (grapes, apples, and coconuts) and grains (rice and wheat). Furthermore, the process is hosted by biological agents, such as fungi and bacteria [92]. Among these are acetic acid bacteria (BAA) and the fungus *Micoderma aceti*, which are characterized by their versatility with regard to the conversion of a series of carbon sources into biomolecules of industrial interest; however, the route of biological synthesis of ethanoic acid represents only 10% of global production [93]. AA can also be obtained from bio-oil that is produced from the pyrolysis of lignocellulosic biomass. In addition, methanol carbonylation, acetaldehyde oxidation, butane/naphtha oxidation, and methyl acetate carbonylation are the main synthetic pathways for producing acetic acid [93].

The use of AA as a reducing agent in the production of rGO promotes the formation of ordered layers with a predominantly clear appearance. The morphological profile presents a slight kneading, an effect that may be responsible for the high surface area and nano structuring properties. During the reduction of a GO produced using the Hummers method, the structure that was initially flocked transforms into a structure with a large number of wrinkles and dents throughout the sheet, structural defects caused due to the high reduction of oxygen in the structure and restructuring of the conjugated structure [94]. AA is commonly combined with salicylic acid as a mixed reducing agent, increasing the level of gratification of the rGO produced when compared to other conventional reducing agents, and is used in the production of nanoparticles; as an additional effect, the dispersity is increased, and the morphological profile by spectroscopy transmission reveals the presence of clear areas without the presence of dents, related to the high reducing effect caused by the process [94]. 

The common reduction process for this hybrid reducer consists of a hydrothermal reduction that assists in the electrical capacity of the rGO. A second method used is vapor reduction that can be used to provide coiled morphologies in order to adhere to some tubular surface, such as threads and fibers. Similarly, the process also helps to reduce the electrical resistance of rGO [95]. The combination of these agents promotes the formation of a material with high electrical conductivity, promoting its use in various sectors. In order to increase the electrical properties of rGOs produced by acetic acid, a binary 3D structure can be formed through vacuum filtration; this ordering tends to increase the charge modulus of the rGO under conditions of low concentrations [96].

#### 1.2.10. Betanin

Betanin (Figure 11) is a pigment belonging to the class of betalains (group of betacyanins), water-soluble nitrogenous pigments widely found in plants of the order Caryophyllales. Betalamic acid is the common precursor of these phenolic compounds, being synthesized from the secondary metabolism of plants and stored mainly in the vacuole of cells [2]. It is considered an excellent antioxidant, acting as a free radical scavenger and an inducer of the antioxidant defense mechanism in cultured cells. Studies indicate its effectiveness in anti-inflammatory, hematoprotective, chemopreventive, and anticancer actions. The substance is found in the root of red beetroot (*Beta vulgaris* L.), pitaya peel (*Hylocereus polyrhizus*), quinoa (*Chenopodium quinoa* Willd), cactus fruits (*Opuntia ficus-indica*), chard (*Beta vulgaris* L. ssp. cicla [L.] Alef. Cv. Bright Lights), and uluco (*Ullucus tuberosus*) [97]. Betanin is considered a promising alternative as a green reducing agent, as the pigment induces the reduction of GO through SN2 nucleophilic reactions and thermal elimination [98].

During a betanin reduction process, the reducing agent is initially extracted through a saline phosphate buffer. With the application of these reducers derived from dragon fruit peel, GO sheets change from a transparent structure covered in wrinkles to a morphology that presents a fine structure, with an increase in wrinkles [98]. The rough morphology is highly affected according to the absorption bands of its absorption spectrum, especially the D (1360 cm^−1^) and G (1590 cm^−1^) bands, which when related due to their Id/Ig intensities, a deoxygenation rate and an average reduction of the sp2 carbon structure are obtained. Increasing this ratio implies an increase in GO reduction, causing a progressive increase in the amount of wrinkle morphology on the surface [99].

The reduction time also implies additional variations in morphology, knowing that an increase in the Id/Ig ratio intensifies the formation of C=C bonds, with a high reduction time causing an extremely rough structure to form flaked structures [100]. This differentiated process led to variations in the sample’s absorption peaks, indicating the restoration of the electronic structure; however, it also presented a reduction in the Id/Ig index, leading to a theoretically smaller reduction. The morphology in this zone showed the formation of laminated aggregates in the rGO morphology, with high thickness, indicative of an inferior reduction quality caused by the excessive acidity of graphene oxide [99,100].

## 2. Perspectives

The growing interest in natural antioxidant agents as a strategy for reducing graphene oxide has been highlighted prominently in the scientific sphere, driven by the potential for biological and environmental applications. It is important to elucidate that these antioxidants, widely available in natural sources such as plants, fruits, and spices, exhibit chemical properties that allow them to interact with and neutralize free radicals as well as other reactive oxygen species associated with graphene oxide. Therefore, the use of compounds such as vitamins C and E, polyphenols, and flavonoids to alter or minimize the reactivity of graphene oxide not only offers a more ecological and safer alternative compared to traditional chemical reducers, but also fosters the development of new techniques for the synthesis and manipulation of graphene-based materials in a sustainable and environmentally friendly manner.

## 3. Conclusions

Reviewing the antioxidant properties of different natural products in reducing graphene oxide is a promising and important area of research. Graphene oxide is an oxidized form of graphene that can cause damage to cells, but the combination of natural antioxidant substances can help protect cells from the harmful effects of GO. This could have significant implications for human health, especially in relation to diseases related to oxidative stress, such as cancer and cardiovascular disease. However, more research is needed to fully understand how these compounds interact and how they can be applied for therapeutic use. Still, the current research is an important step toward advancing our knowledge about the properties of graphene and how we can use them to improve human health and well-being.

## Figures and Tables

**Figure 1 ijms-25-05182-f001:**
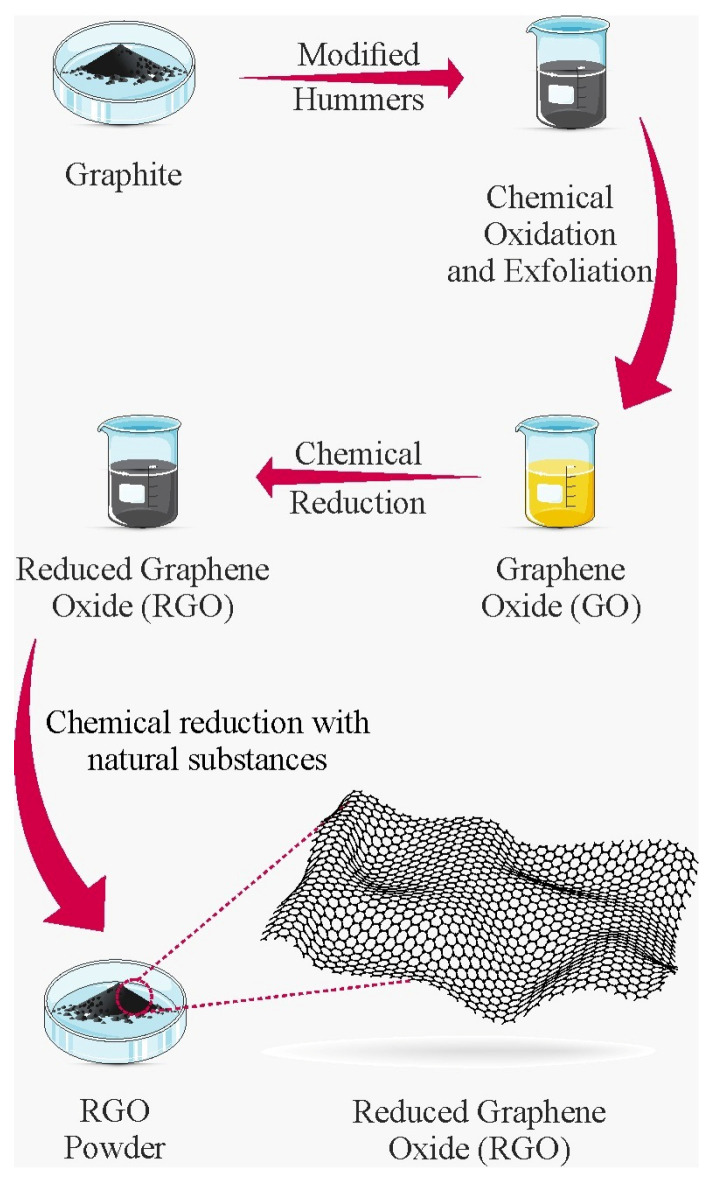
Chemical reduction of GO. Elaborated by the authors.

**Figure 2 ijms-25-05182-f002:**
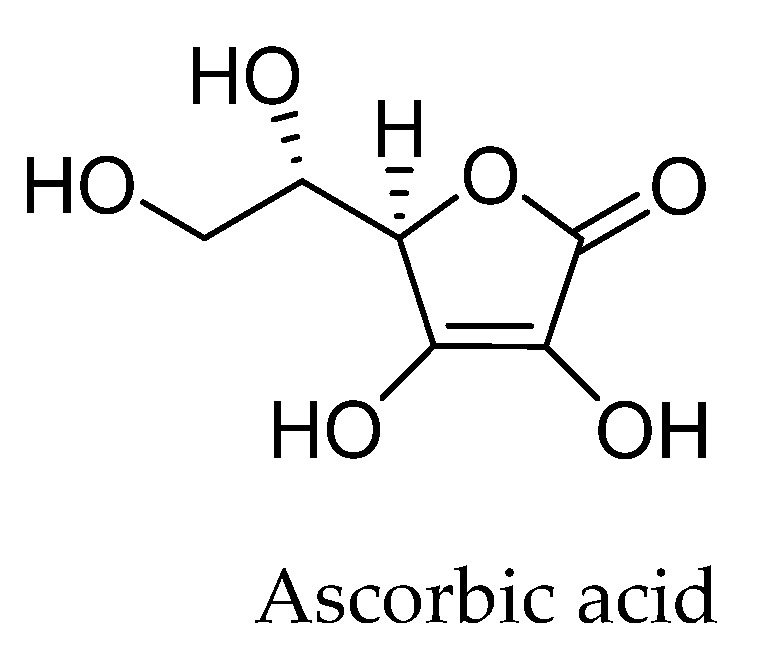
Chemical structure of ascorbic acid. Elaborated by the authors.

**Figure 3 ijms-25-05182-f003:**
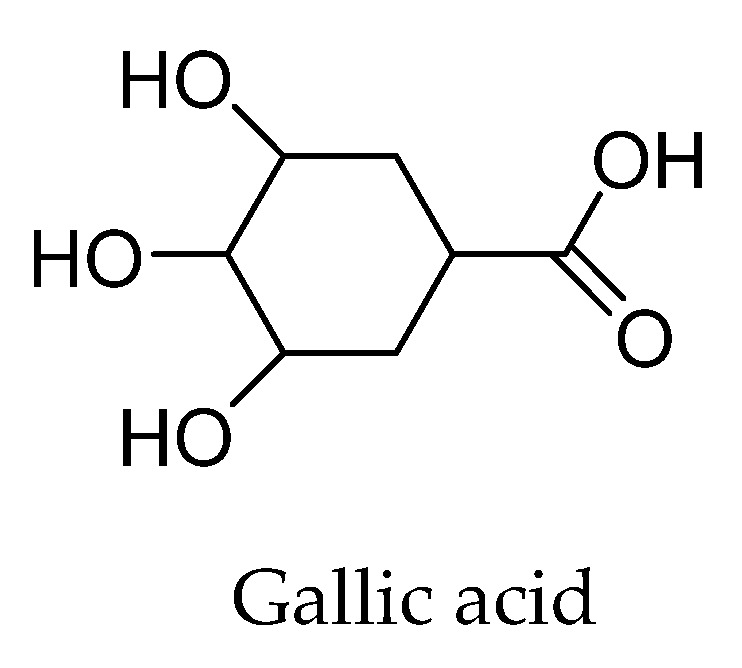
Chemical structure of gallic acid. Elaborated by the authors.

**Figure 4 ijms-25-05182-f004:**
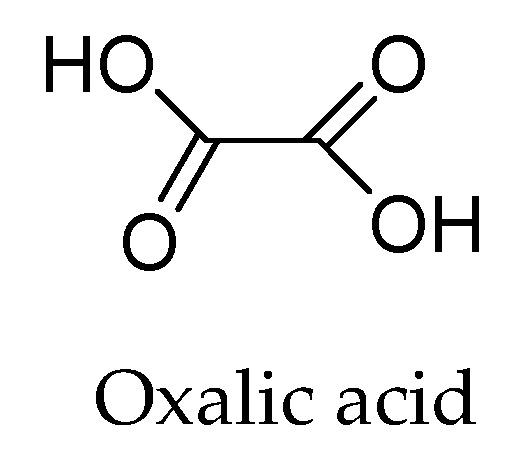
Chemical structure of oxalic acid. Elaborated by the authors.

**Figure 5 ijms-25-05182-f005:**
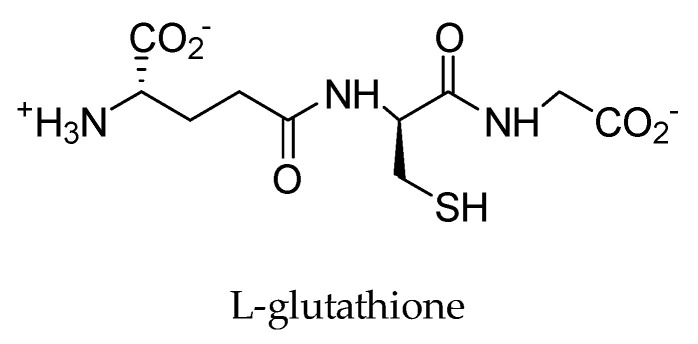
Chemical structure of L-glutathione. Elaborated by the authors.

**Figure 6 ijms-25-05182-f006:**
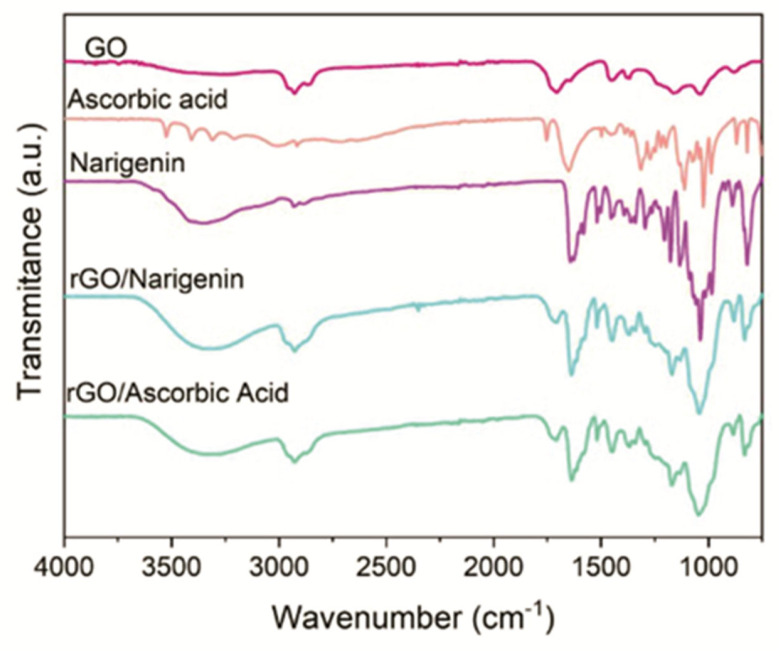
FTIR spectrum of GO, graphene, and other elements. Elaborated by the authors.

**Figure 7 ijms-25-05182-f007:**
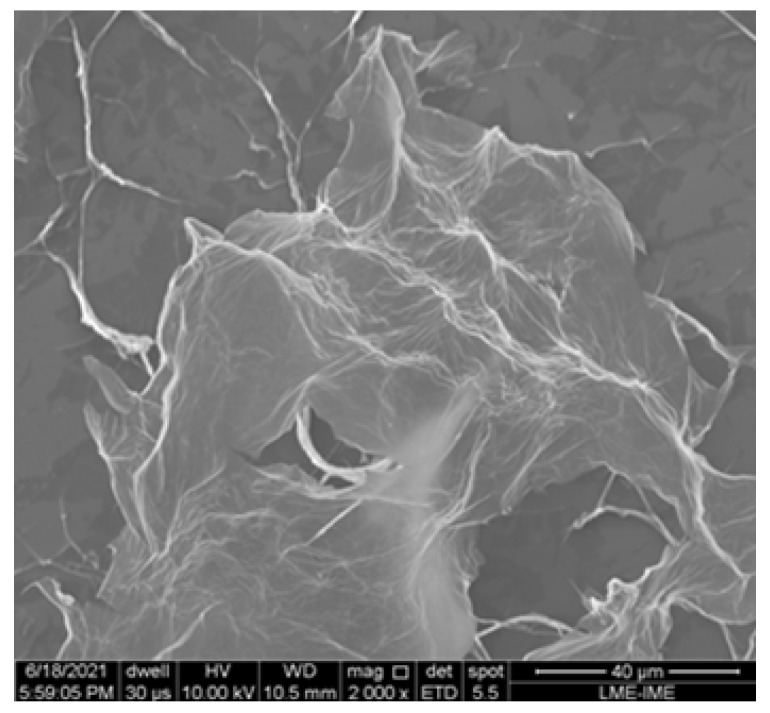
Morphology of GO. Elaborated by the authors.

**Figure 8 ijms-25-05182-f008:**
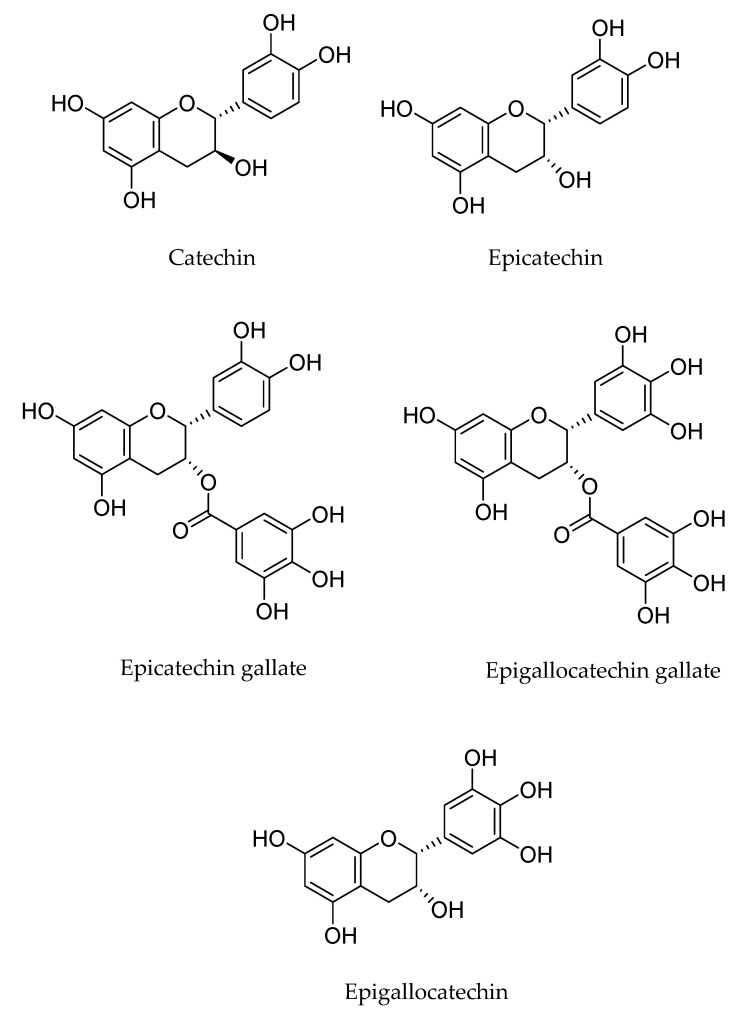
Main polyphenols in green tea. Elaborated by the authors.

**Figure 9 ijms-25-05182-f009:**
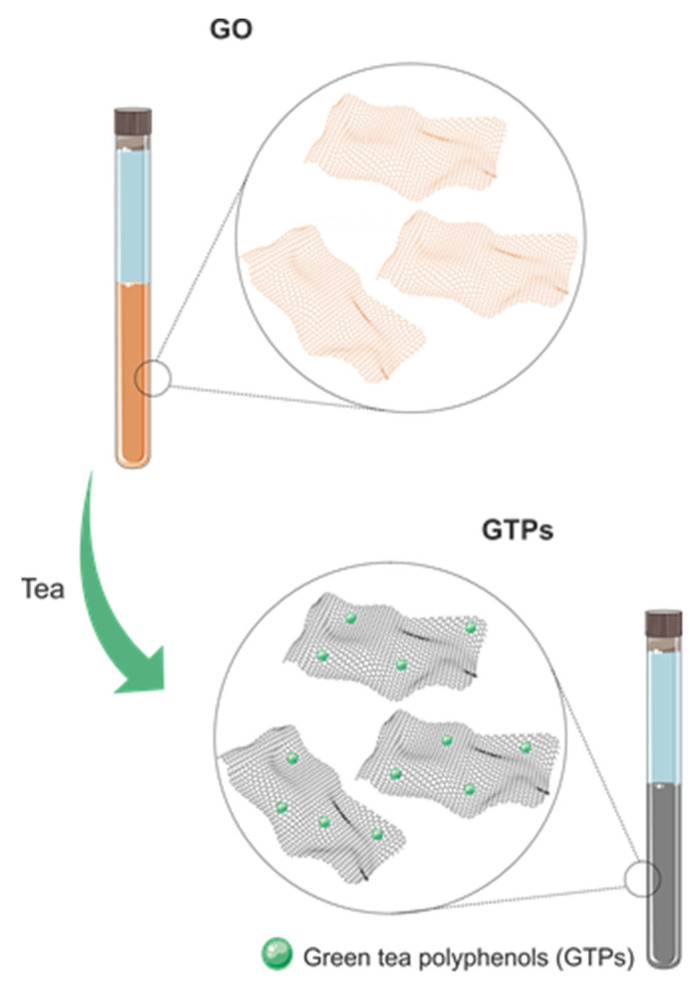
Illustration of process of reduction of GO with green tea polyphenols (GTPs). Elaborated by the authors.

**Figure 10 ijms-25-05182-f010:**
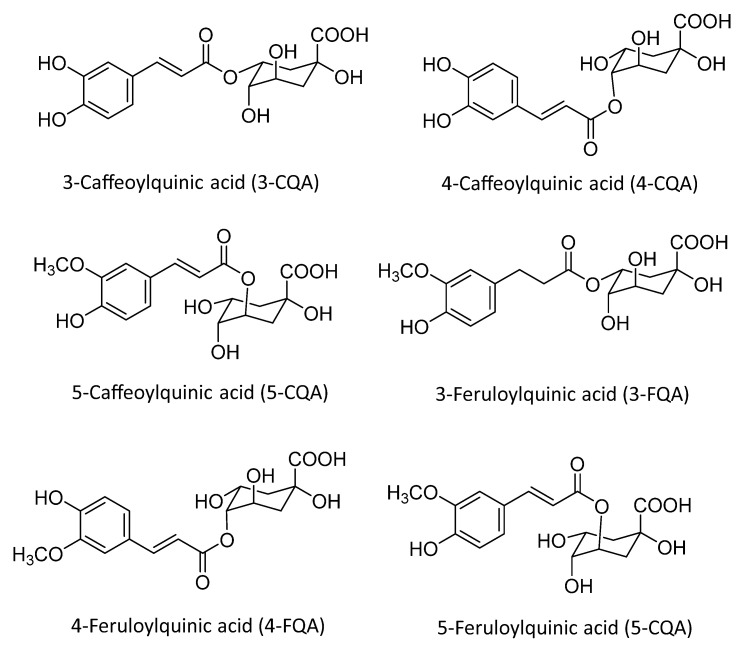
Main hydroxycinnamic acids described in coffee. Elaborated by the authors.

**Figure 11 ijms-25-05182-f011:**
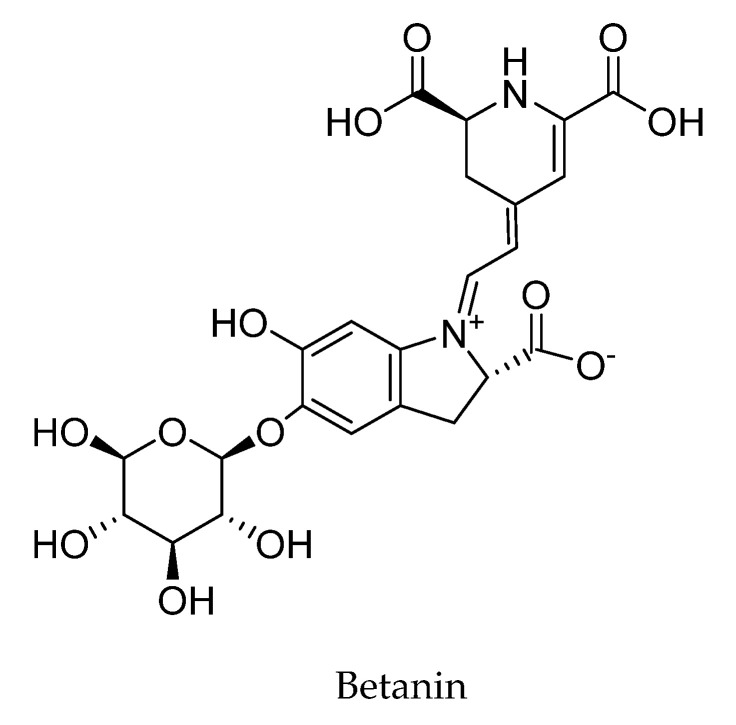
Chemical structure of betanin. Elaborated by the authors.

**Table 1 ijms-25-05182-t001:** Examples of the application of ascorbic acid as a reducing agent.

Ref.	Sample	Reduction Method	Reduction Temperature	Equipment	Observations	[AA]	[GO]
[1]	Graphite powder	Chemical	70 °C, 120 °C, and 300 °C	XRD, XPS, UV-Vis-NIR	Optical performance analysis of rGO thin films with AA comparing them with hydrazine at the same concentrations.	5, 25, 50, 100, 200, 400, and 800 mM	10 mg/mL
[3]	Graphite flakes	Chemical	90 °C/24 h	XRD, XPS, FE-SEM, DLS	Structural, electrical, and optical analysis of rGO films with AA.	30 mg/mL	32 and 100 mg/mL
[4]	Monolayer graphene	Chemical	70 °C/5 min	SEM, Raman, FTIR, XRD	Production of centimeter-scale semipermeable membranes with ion blocking efficiency of up to 9.	0.4 g/mL	4 mg/mL
[34]	Natural graphite	Chemical	Room temperature	UV-Vis-NIR, FT-IR, XPS	AA as photosensitive reductant of GO	5 mg/mL	0.1 mg/mL
[6]	Natural graphite	Chemical	65 °C/1 h	DRX, XPS, SEM, AFM, ATR-FT-IR	Characterization of reduced GO by AA in different periods	100 mg/mL	0.1 mg/mL
[7]	Multilayer GO	Hydrothermal	95 °C/6 h	SEM, XRD, FT-IR	Fabrication of three-dimensional reduced GO aerogels (rGOAs) using AA as a reductant	240 mg/mL	4.0 mg/mL
[28]	Expanded graphite	Hydrothermal	80 °C/24 h	XPS, Raman, TEM, SEM	Hydrogel with piezoresistive property manufactured with GO reduced with Pt and AA nanoparticles	10 mg/mL	5, 10, and 15 mg/mL

**Table 2 ijms-25-05182-t002:** Examples of the application of oxalic acid as a reducing agent.

Ref.	Sample	Reduction Method	Reduction Temperature	Equipment	Observations	[OA]	[GO]
[35]	Graphite powder	Chemical	75 °C/18 h	TEM, SEM, FT-IR, XRD	Synthesis of graphene nanosheets via oxalic acid-induced chemical reduction	75.6 mg/mL	2.5 mg/mL
[36]	Natural graphite	Chemical	900 °C/2 h	XRD	Exfoliation and reduction method	10 mg/mL	2 mg/mL
[29]	Natural graphite powder	Hydrothermal	160 °C/5 min	SEM, Raman, FTIR, XRD	Production of centimeter-scale semipermeable membranes with ion blocking efficiency of up to 90%	6, 12, and 25 g/mL	1.5 mg/mL
[33]	Synthetic graphite	Hydrothermal	100 °C/1 h	DRX, FT-IR, SEM, XPS	Composite of rGO and bismuth as electrode for super capacitor	18 mg/mL	1.5 mg/mL
[37]	GO	Hydrothermal Chemical	180 °C/24 h	UV-Vis-NIR, FT-IR, XPS	AA as photosensitive reductant of GO	4.4 mg/mL	0.8 mg/mL
[38]	Natural graphite	Hydrothermal	200 °C/12 h	DRX, XPS, SEM, AFM, ATR-FT-IR	Characterization of reduced GO, by AA in different periods	NI	NI
[39]	Multilayer GO	Chemical	120 °C/13 h	SEM, XRD, FT-IR	Fabrication of three-dimensional reduced GO aerogels (rGOAs) using AA as a reductant	NI	NI
[40]	Expanded graphite	Hydrothermal	50 °C/12 h	XPS, Raman, TEM, SEM	Hydrogel with piezoresistive property manufactured with GO reduced with Pt and AA nanoparticles	NI	NI

**Table 3 ijms-25-05182-t003:** Examples of the application of gallic acid as a reducing agent.

Ref.	Sample	Reduction Method	Reduction Temperature	Equipment	Observations	[GA]	[GO]
[1]	Graphite powder	Hydrothermal	50 °C/12 h	FT-IR, SEM	Nanoformulation based on graphene oxide loaded with gallic acid (GAGO)	0.25 g/mL	0.05 g/mL
[3]	Graphite powder	Chemical	45 °C/2 h	FT-IR, TEM, XRD, DLS	Green synthesis of metal nanoparticles	10 mL of 2 mM GA	3.4 mL of 1 mg/mL GO
[4]	Graphite powder	Chemical	Room temperature 24 h	UV-Vis, IR, Raman, XPS	Gallic acid as a GO reducer and stabilizer	4000 mg/mL	4000 mg/mL
[7]	Graphite powder	Chemical	100 °C/8 h	FESEM, XPS, XRD, TGA, UV-Vis DRS	Graphene hydrogel with gallic acid as an adsorbent of wastewater pollutants	2 mg/mL	2 mg/mL
[28]	GO	Hydrothermal	190 °C/5 h	TEM, DRX, XPS	Ag nanoparticles (AgNP) are added to rGO sheets using gallic acid-capped AuNPs.	5 mg/mL	2.5 mg/mL
[29]	Carbon nanotubes	Chemical	Room temperature	UV-Vis TEM, FT-IR, FT-Raman	MWCNT-rGO nanocomposite electrode for sensitive detection of gallic acid capped AuNPs	10 mM and 100 mM	0.2 mg/mL
[40]	Multilayer GO	Chemical	80 °C/24 h	FE-SEM, TEM, FT-IR, XRD, XPS	Graphene nanotube hybrid papers for high-performance electrochemical capacitive energy storage	480 mg/mL	0.1 mg/mL

**Table 4 ijms-25-05182-t004:** Examples of the application of GSH as a reducing agent.

Ref.	Sample	Reduction Method	Reduction Temperature	Equipment	Observations	[GSH]	[GO]
[41]	Natural graphite powder	Chemical	50 °C/6 h	TEM, SEM, FT-IR, XRD	Preparation of graphene nanosheets from GO and l-glutathione	2 mg/mL	0.1 mg/mL
[42]	Natural graphite flakes	Chemical	50 °C/5 h	FE-SEM, FT-IR, XRD, TGA	L-glutathione modified graphene/epoxy composites	1200 mg/mL	400 mg/mL
[43]	Natural graphite powder	Chemical	60 °C/2 h	XRD, FT-IR, XPS, FE-SEM, EIS, DPV	GO/ZnO nanocomposite with glutathione as electrode for piroxicam sensor	1 g/mL	1 mg/mL
[44]	Synthetic graphite	Chemical	Room temperature	EIS, SEM, EDX	Composite of rGO and bismuth as electrode for super capacitor	0.03 mg/mL	0.05 mg/mL

**Table 5 ijms-25-05182-t005:** Examples of the application of polyphenols as a reducing agent.

Ref.	Sample	Reduction Method	Reduction Temperature	Equipment	Observations	[Polyphenol]	[GO]
[45]	Natural graphite powder	Chemical	80 °C/8 h	FE-SEM, FT-IR, UV-Vis, XRD	High-efficiency anticancer drug delivery systems using functionalized RGO tea polyphenols	NI	500 mg/L
[46]	Natural graphite flakes	Chemical	Room temperature	XPS, AFM, Raman	Antioxidant activity of green tea polyphenols in the presence of iron for the reduction of GO	100 mg/L	100 mg/L
[47]	Natural graphite powder	Chemical	90 °C/8 h	Uv-Vis, FT-IR, XPS, TGA	Mechanism of rGO reduction and cytotoxicity by green tea polyphenol (GPT-rGO)	NI	300 mg/L
[38]	Synthetic graphite	Chemical	Room temperature/24 h	FE-SEM, TEM, EIS, SWASV, XPS	Tea polyphenol-mediated rGO/iron nanocomposite for electrochemical determination of Hg^2+^	5000 mg/mL	10,000 mg/mL
[39]	GO powder	Chemical	80 °C/2 h	Uv-Vis, TEM, FEL, WXRD, FT-IR, XPS	In situ synthesis of Ag nanoparticles in GO modified with tea polyphenols	2835 mg/L	500 mg/L

**Table 6 ijms-25-05182-t006:** Examples of glucose application as a reducing agent.

Ref.	Sample	Reduction Method	Reduction Temperature	Equipment	Observations	[Glucose]	[GO]
[47]	GO	Chemical	95 °C/1 h	TEM, FTIR, XRD, TGA	Synthesis of chemically converted GNS based on reducing sugars and GO as precursor	0.1 mg/mL	0.1 mg/L
[40]	Natural graphite	Chemical	95 °C/30 min	AFM, UV-Vis-NIR, XPS	Glucose-reduced GO sheets in photothermal therapy of prostate cancer cells	100 μL	1 and 0.05 mg/L
[1]	Pure graphite	Chemical	Room temperature/24 h	Uv-Vis, FT-IR, XPS, EDX, XRD	Ag-Cu_2_O nanocomposites supported on GO with glucose reduction (rGO) and visible light photocatalytic activity	15 mL	50 mg/L
[32]	Graphite flakes	Chemical	135 °C/30 min	AFM, FT-IR, UV-Vis, NIR	rGO for glucose excellent biocompatibility and photothermal efficiency and drug loading	22 mg/mL	1 mg/mL
[33]	GO	Hydrothermal	160 °C/12 h	XRD, XPS, FT-IR, UV-Vis, Raman, SEM, and DLS	GO rich in carbonyl functional groups partially reduced using glucose as the only precursor	50 mL	NI

## Data Availability

Data are available on request.

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
