# Peer review of "Graphene and Natural Products: A Review of Antioxidant Properties in Graphene Oxide Reduction"

_ijms, 2024, doi:10.3390/ijms25105182_

Round 1
Reviewer 1 Report
Comments and Suggestions for Authors
Through this review article, the authors address extremely interesting issues referred to antioxidant properties of different natural products in relation to the reduction of graphene oxide. Their survey clearly demonstrates that using these antioxidant compounds can be potentially an effective strategy to minimize the harmful impact of graphene oxide on living cells. I strongly recommend publication of this worthwhile contribution in International Journal of Molecular Sciences and have only few minor suggestions to polish the presentation of this interesting material:
Point #1
The authors rightly note that antioxidant compounds they consider are able to protect living cells from oxidative stress. In this context, it is noteworthy that there exist the two major mechanisms for oxidative stress-induced biological consequences. The first mechanism involves the reactive oxygen species (ROS) and reactive nitrogen species (RNS) generation, which are responsible for the oxidation of biomolecules leading to cellular dysfunctions and death, while the second oxidative-stress mechanism involves aberrant redox signaling. Accordingly, antioxidants may act as ROS and RNS scavengers [1,2] or through the impact on redox signaling [3]. Thus, I would suggest to mention this circumstance along with the given references and provide at least an idea on whether the reactivity of the considered compounds towards reduction of graphene oxide correlates better with one or another facet of antioxidant activity (i.e. radical scavenging versus influencing the redox signaling).
Additional references:
1. R.F. Vasil’ev et al., Kinetics and Catalysis 2014, 55, 148-153.
2. G.F. Fedorova et al., Photochemistry and Photobiology 2017, 93, 579-589.
3. L. Flohé, Antioxidants 2020, 9, 1254.
Point #2
I would provide the pertinent literature reference(s) to the first two sentences of subsection 1.3.6 (lines 259-262).
Point #3
Subsections 1.3.5. and 1.3.11 are about the same compound and bear the same title (Bacteriorhodopsin). How about combining these subsections?
Author Response
Dear Editor,
We would like to express our sincere appreciation for your swift attention to our manuscript, IJMS-2952302, titled "Graphene and natural products: a review of antioxidant prop-erties in graphene oxide reduction". It is truly gratifying to witness a journal that prioritizes efficiency and expedience in its review process.
We extend our sincere gratitude to the reviewers for their meticulous analysis and constructive feedback. Their contributions have undeniably elevated the quality and clarity of our manuscript. We have carefully considered and addressed all of their comments, ensuring that each point is either attended to or justified. Below, we provide a comprehensive list of the changes made, with corresponding references to the main text where these modifications can be observed, highlighted for clarity.
With my very best regards,
Filipe.
Reviewer 1
Point #1
Answer: The introduction of reduced graphene oxide (rGO) into cells can induce oxidative stress. In the context of oxidative stress-related biological outcomes, two primary mechanisms dominate. The first mechanism centers on the production of reactive oxygen species (ROS) and reactive nitrogen species (RNS), leading to biomolecular oxidation, consequentially resulting in cellular dysfunction and mortality. Simultaneously, the second oxidative stress mechanism revolves around the perturbation of redox signaling pathways. Consequently, antioxidants play a crucial role, either by scavenging ROS and RNS [8,9] or by modulating redox signaling pathways [10].
As recommended, the following references were inserted:
[14] Vasil’ev, R. F., Veprintsev, T. L., Dolmatova, L. S., Naumov, V. V., Trofimov, A. V., & Tsaplev, Y. B. (2014). Kinetics of ethylbenzene oxy-chemiluminescence in the presence of antioxidants from tissues of the marine invertebrate Eupentacta fraudatrix: Estimating the concentration and reactivity of the natural antioxidants. Kinet. Catal., 55, 148-153. DOI: 10.1134/S0023158414020153
[15] Fedorova, G. F., Menshov, V. A., Trofimov, A. V., Tsaplev, Y. B., Vasil'ev, R. F., & Yablonskaya, O. I. (2017). Chemiluminescence of cigarette smoke: Salient features of the phenomenon. J. Photochem. Photobiol., A, 93(2), 579-589. DOI: 10.1111/php.12689
[16] Flohé, L. (2020). Looking back at the early stages of redox biology. J. Antioxid. Act., 9(12), 1254. DOI: 10.3390/antiox9121254
Point #2
Answer: As recommended, the following references were inserted:
[54] Tielan Wei, Sachin Sunil Thakur and Mengyang Liu et al. Oral delivery of glutathione: antioxidant function, barriers and strategies. Acta Materia Medica. 2022. Vol. 1(2):177-192. DOI: 10.15212/AMM-2022-0005
[56] Barbosa, D.S. Green Tea Polyphenolic Compounds and Human Health. J. Verbr. Lebensm. 2, 407–413 (2007). https://doi.org/10.1007/s00003-007-0246-z
[57] Yan Z, Zhong Y, Duan Y, Chen Q, Li F. Antioxidant mechanism of tea polyphenols and its impact on health benefits. Animal Nutrition. 2020 Jun 1;6(2):115-23. https://doi.org/10.1016/j.aninu.2020.01.001.
Point #3
Answer: The term was removed as solicited.

Reviewer 2 Report
Comments and Suggestions for Authors
- The significance of nanomaterials in the management of diseases should be discussed in the introduction section.
- The authors should elaborately discuss the applications of graphene and its derivatives in the introduction section.
- The authors discussed many natural antioxidants, such as ascorbic acid, etc., but they did not mention the most powerful antioxidant among the discussed antioxidants.
- The authors should provide the data for the toxicity of graphene oxide prepared by chemical reduction methods with references.
- Issues and challenges in the green reduction of graphene oxide should be discussed in a new paragraph.
- There are several limitations to using green reducing agents. The authors did not discuss.
- Please make a new paragraph about the perspective.
- What is the novelty of your work compared to other research?
- Improve the grammatical and syntax errors.
- Improve the grammatical and syntax errors.
Author Response
Dear Editor,
We would like to express our sincere appreciation for your swift attention to our manuscript, IJMS-2952302, titled "Graphene and natural products: a review of antioxidant prop-erties in graphene oxide reduction". It is truly gratifying to witness a journal that prioritizes efficiency and expedience in its review process.
We extend our sincere gratitude to the reviewers for their meticulous analysis and constructive feedback. Their contributions have undeniably elevated the quality and clarity of our manuscript. We have carefully considered and addressed all of their comments, ensuring that each point is either attended to or justified. Below, we provide a comprehensive list of the changes made, with corresponding references to the main text where these modifications can be observed, highlighted for clarity.
With my very best regards,
Filipe.
Reviewer 2
The significance of nanomaterials in the management of diseases should be discussed in the introduction section.
The authors should elaborately discuss the applications of graphene and its derivatives in the introduction section.
Answer: We appreciate the reviewer's suggestion, very important. In electronics, graphene's high electrical conductivity and transparency have led to advancements in flexible and transparent electronic devices. Moreover, in energy storage, graphene-based materials show potential for enhancing the performance of supercapacitors and batteries due to their large surface area and excellent conductivity [8,9]. In the realm of medical science, the exploration of nanomaterials (as graphene and its derivatives) has emerged as a groundbreaking avenue in disease management, offering a promising application in healthcare. By graphene's unique characteristics, researchers are developing advanced drug delivery systems capable of precisely targeting diseased cells while minimizing collateral damage to healthy tissues [10]. Furthermore, graphene-based biosensors exhibit exceptional sensitivity, enabling early detection and monitoring of various diseases, from cancer to neurological disorders [11,12]. Additionally, graphene's antimicrobial properties hold a potential in combating antibiotic-resistant pathogens, presenting a promising solution to the global healthcare challenge of infectious diseases [13].
The authors discussed many natural antioxidants, such as ascorbic acid, etc., but they did not mention the most powerful antioxidant among the discussed antioxidants.
Answer: Ascorbic acid, or vitamin C, is recognized as a highly effective reductant for converting graphene oxide to graphene, a process supported by scientific evidence highlighting its remarkable properties. With a reduction potential of +0.06 V against the Standard Hydrogen Scale at pH 7, ascorbic acid efficiently donates electrons, reverting graphene oxide to graphene with improved conductive and mechanical properties. Its high solubility in water promotes effective interaction and homogeneous dispersion in aqueous solutions, facilitating a complete reduction that results in high-quality graphene with fewer structural defects. Furthermore, its lower toxicity and greater biocompatibility compared to other common reductants such as hydrazine or sodium borohydride make ascorbic acid safer and more sustainable for applications in biotechnology and other areas. Its wide availability, stability and antioxidant properties also contribute to its effectiveness and practicality, protecting the formed graphene from oxidative degradation. Therefore, ascorbic acid not only proves to be an efficient reducing agent due to its physicochemical characteristics, but also due to its environmental and safety advantages, making it an ideal choice for the sustainable production of graphene in scientific and industrial applications.
The authors should provide the data for the toxicity of graphene oxide prepared by chemical reduction methods with references.
Answer: On the other hand, the method of obtaining rGO through different synthesis can be worrying due to the risk of toxicity, such as reduction with hydrazine [17]. Another point to be highlighted is that GO has been shown to be a biocompatible material, however, when inserted into cell culture medium, there is a loss of cell functionality in 20% and 50% when added 20 mg.ml-1 and 50 mg.mg-1 respectively [18]. Furthermore, studies reports that the presence of graphene-based particles in the blood can travel to different organs in our body and can reach the nervous system, and when ingested intravenously or orally, they reach mainly the kidneys, lungs and liver, leading to inflammation [19,20]. Another author with in vivo studies shows that prolonged exposure to GO in doses of 25 micrograms L-1 caused opacity in the animals' eyeballs, in contrast to RGO, which exhibited biosafety [21].
Issues and challenges in the green reduction of graphene oxide should be discussed in a new paragraph.
Answer: Concern about environmental issues, especially in terms of minimizing or eliminating harmful chemical substances or those that result in toxic waste, has drawn the attention of many researchers to obtain reduced graphene oxide, and this method has been called green reduction of graphene oxide. Among the various approaches, one can include the use of natural reducing agents, such as plant extracts, and reduction methods using water as a solvent. However, the big challenge of this green route is related to efficiency when compared to the synthesis route with hydrazine, for example, and obtaining an rGO with good properties. Among all the green routes mentioned by De Silva et. al., the one that showed the most promise was the one used with ascorbic acid to obtain RGO. In addition to being a recent process, it has shown excellent results compared to hygdrazine methods, however, more research is needed in order to model and find a better optimization process for this route with changing parameters and conditions in search of large-scale production [22-24].
There are several limitations to using green reducing agents. The authors did not discuss.
Answer:
Please make a new paragraph about the perspective.
Answer: The growing interest in natural antioxidant agents as a strategy for reducing graphene oxide has been highlighted prominently in the scientific sphere, driven by the potential for biological and environmental applications. It is important to elucidate that these antioxidants, widely available in natural sources such as plants, fruits, and spices, exhibit chemical properties that allow them to interact with and neutralize free radicals as well as other reactive oxygen species associated with graphene oxide. Therefore, the use of compounds such as vitamins C and E, polyphenols, and flavonoids to alter or minimize the reactivity of graphene oxide not only offers a more ecological and safer alternative compared to traditional chemical reducers, but also fosters the development of new techniques for the synthesis and manipulation of graphene-based materials in a sustainable and environmentally friendly manner.
What is the novelty of your work compared to other research?
Answer: This review article presents a detailed and comprehensive analysis on the potential applications of natural antioxidants in the graphene oxide reduction process. The review stands out for its depth and scientific rigor, offering a solid foundation of knowledge that not only expands understanding of reduced graphene oxide, but also sheds light on the effective use of natural reducing agents. Furthermore, it appears that this study also investigates the environmental and biological implications of these interactions, suggesting promising paths for future research and technological applications that combine sustainability and innovation in materials engineering.

Reviewer 3 Report
Comments and Suggestions for Authors
Although the work is interesting, interestingly conceived and well presented (two schemes have been specially designed, Figs. 1 and 2), there are some strange errors that cast a shadow on the quality of the research itself.
Which chemistry textbook says that glycogen consists of glucose and fructose? Even common carbohydrate monomers are not glucose and fructose. Please, rewrite the part (line 297-305) with exact and correct facts.
Incorrect statement that sucrose is a reducing sugar. Moreover, is sucrose a part of lignocellulose? (line 352)
What is oxyrdic acid? (line 422).
There is only one figure 5, and the authors refer to figures 5a and 5b (lines 226 and 233).
A very strange formulation of "Figure 10" is used... as if it were part of this work, but ultimately it is a citation, i.e. an explanation using the cited work? If so, it should be worded more clearly. (line 255).
The first sentence, which explains etanoic acid, is neither clear nor precise (lines 396 /397).
It would make sense to divide the compounds in Figure 3 according to the chapters in which they occur.
Minor mistakes: Ccorrect name for caffeic acid is : (E)-3-(3,4-dihydroxyphenyl)prop-2-enoic acid (line 361)
or
- 3,4-dihydroxycinnamic acid (line 360).
what type of mistake is stress49? (line 367).
you use abbreviation CA for caffeic acid, If so, correct line 374.
Table 2 - wrong abbreviation for oxalic acid (in the head line).
Table 3, 4 and 5 - head line: Equipaments
Lines 116-118 - sentence not clear (due to used twice), please rewrite.
Lines 183-186 - sentence with missing ending?
Line 447. betatin instead betanin
line 457 prGOressive
Lines 461 - 467. Figure 9 made a part of some citation, or? it is neither clearly written nor explained.
Comments on the Quality of English Language
There are some sentences that are not clear enough (already mentioned in the previous part)
Author Response
Reviewer 3
Although the work is interesting, interestingly conceived and well presented (two schemes have been specially designed, Figs. 1 and 2), there are some strange errors that cast a shadow on the quality of the research itself.
Answer: We appreciate the reviewer's suggestion, very important. The figures cited were adjusted, one of them being removed for better understanding of the text.
Which chemistry textbook says that glycogen consists of glucose and fructose? Even common carbohydrate monomers are not glucose and fructose. Please, rewrite the part (line 297-305) with exact and correct facts.
Answer: We thank the reviewer for the commentary. The excerpt was adjusted with reliable information and their respective references. According to the literature, carbohydrates, also called glycans, are one of the most prevalent biopolymers in nature, sharing the role of three essential molecules of life alongside polynucleotides and polypeptides. Its presence is ubiquitous, found in all liv-ing beings, from animals and plants to microbes. Furthermore, its function is to play significant roles in physiological processes. Basically, the carbohydrate molecule is composed of carbon, hydrogen and oxygen atoms, having a general chemical formula (CH2O)n. In this sense, monosaccharides such as glucose (GI) are important sugars for the energy metabolism of most living organisms. Furthermore, is commercially availa-ble in the form of a white, crystallized, sweet and odorless powder, has the formula C6H12O6 and molecular weight 180.16 g/mol.
Incorrect statement that sucrose is a reducing sugar. Moreover, is sucrose a part of lignocellulose?
Answer: The sentence was corrected. Sugarcane bagasse is mainly composed of hemicellulose (25–35%), lignin (15–35%) and cellulose (40–45%), and thermochemical treatment breaks the lignocellulosic bonds and partially solubilizes the polysaccharides to release sugars, such as glucose and sucrose.
What is oxyrdic acid?
Answer: The error has been corrected.
There is only one figure 5, and the authors refer to figures 5a and 5b.
Answer: The error has been corrected.
A very strange formulation of "Figure 10" is used... as if it were part of this work, but ultimately it is a citation, i.e. an explanation using the cited work? If so, it should be worded more clearly.
Answer: The error has been corrected.
The first sentence, which explains etanoic acid, is neither clear nor precise
Answer: We thank the reviewer for the commentary. The sentence was corrected. Ethanoic acid (CH₃COOH) or acetic acid (AA) is an organic compound with a saturat-ed and open chain, it is an important chemical product widely used in the food industry.
It would make sense to divide the compounds in Figure 3 according to the chapters in which they occur.
Answer: The figures were distributed according to the topics presented.
Minor mistakes: Ccorrect name for caffeic acid is : (E)-3-(3,4-dihydroxyphenyl)prop-2-enoic acid
Answer: The error has been corrected.
What type of mistake is stress49?
Answer: The error has been corrected.
You use abbreviation CA for caffeic acid, If so, correct line
Answer: The error has been corrected.
Table 2 - wrong abbreviation for oxalic acid
Answer: The error has been corrected.
Table 3, 4 and 5 - head line: Equipaments
Answer: The error has been corrected.
Lines 116-118 - sentence not clear (due to used twice), please rewrite.
Answer: The sentence was corrected. Polyphenols (Table 5) and Glucose (Table 6) are natural reducers whose use has grown in recent years, this is due to their characteristics and applications, such as drug deliv-ery systems and treatments for specific conditions of the human body.
Lines 183-186 - sentence with missing ending?
Answer: The sentence was corrected. Based on previous studies, rGO reduced using oxalic acid (rGO/AO) as red agent, has a rough surface, folded edges, ripples, multilayer structure, tangles of graphene layers and presence of ultra-thin and transparent graphene sheets as profile characteristic morphological. Furthermore, in view of this, it is noteworthy that the folds in the edge regions confer high tenacity to the material, therefore leading to excellent mechanical properties.
Line 447. betatin instead betanin
Answer: The error has been corrected.
Line 457 prGOressive
Answer: The error has been corrected.
Lines 461 - 467. Figure 9 made a part of some citation, or? it is neither clearly written nor explained.
Answer: The sequence of figures has been adjusted, with correct citation throughout the text.

Round 2
Reviewer 3 Report
Comments and Suggestions for Authors
Sorry, but instead of the new version of the manuscript there is the authorship change form.
I would really like to see the final version to make my final answer/decision.
Author Response
Dear Editor,
We would like to express our sincere appreciation for your swift attention to our manuscript, IJMS-2952302, titled "Graphene and natural products: a review of antioxidant prop-erties in graphene oxide reduction". It is truly gratifying to witness a journal that prioritizes efficiency and expedience in its review process.
We extend our sincere gratitude to the reviewers for their meticulous analysis and constructive feedback. Their contributions have undeniably elevated the quality and clarity of our manuscript. We have carefully considered and addressed all of their comments, ensuring that each point is either attended to or justified. Below, we provide a comprehensive list of the changes made, with corresponding references to the main text where these modifications can be observed, highlighted for clarity.
With my very best regards,
Filipe.
Reviewer 3
Although the work is interesting, interestingly conceived and well presented (two schemes have been specially designed, Figs. 1 and 2), there are some strange errors that cast a shadow on the quality of the research itself.
Answer: We appreciate the reviewer's suggestion, very important. The figures cited were adjusted, one of them being removed for better understanding of the text.
Which chemistry textbook says that glycogen consists of glucose and fructose? Even common carbohydrate monomers are not glucose and fructose. Please, rewrite the part (line 297-305) with exact and correct facts.
Answer: We thank the reviewer for the commentary. The excerpt was adjusted with reliable information and their respective references. According to the literature, carbohydrates, also called glycans, are one of the most prevalent biopolymers in nature, sharing the role of three essential molecules of life alongside polynucleotides and polypeptides. Its presence is ubiquitous, found in all liv-ing beings, from animals and plants to microbes. Furthermore, its function is to play significant roles in physiological processes. Basically, the carbohydrate molecule is composed of carbon, hydrogen and oxygen atoms, having a general chemical formula (CH2O)n. In this sense, monosaccharides such as glucose (GI) are important sugars for the energy metabolism of most living organisms. Furthermore, is commercially availa-ble in the form of a white, crystallized, sweet and odorless powder, has the formula C6H12O6 and molecular weight 180.16 g/mol.
Incorrect statement that sucrose is a reducing sugar. Moreover, is sucrose a part of lignocellulose?
Answer: The sentence was corrected. Sugarcane bagasse is mainly composed of hemicellulose (25–35%), lignin (15–35%) and cellulose (40–45%), and thermochemical treatment breaks the lignocellulosic bonds and partially solubilizes the polysaccharides to release sugars, such as glucose and sucrose.
What is oxyrdic acid?
Answer: The error has been corrected.
There is only one figure 5, and the authors refer to figures 5a and 5b.
Answer: The error has been corrected.
A very strange formulation of "Figure 10" is used... as if it were part of this work, but ultimately it is a citation, i.e. an explanation using the cited work? If so, it should be worded more clearly.
Answer: The error has been corrected.
The first sentence, which explains etanoic acid, is neither clear nor precise
Answer: We thank the reviewer for the commentary. The sentence was corrected. Ethanoic acid (CH₃COOH) or acetic acid (AA) is an organic compound with a saturat-ed and open chain, it is an important chemical product widely used in the food industry.
It would make sense to divide the compounds in Figure 3 according to the chapters in which they occur.
Answer: The figures were distributed according to the topics presented.
Minor mistakes: Ccorrect name for caffeic acid is : (E)-3-(3,4-dihydroxyphenyl)prop-2-enoic acid
Answer: The error has been corrected.
What type of mistake is stress49?
Answer: The error has been corrected.
You use abbreviation CA for caffeic acid, If so, correct line
Answer: The error has been corrected.
Table 2 - wrong abbreviation for oxalic acid
Answer: The error has been corrected.
Table 3, 4 and 5 - head line: Equipaments
Answer: The error has been corrected.
Lines 116-118 - sentence not clear (due to used twice), please rewrite.
Answer: The sentence was corrected. Polyphenols (Table 5) and Glucose (Table 6) are natural reducers whose use has grown in recent years, this is due to their characteristics and applications, such as drug deliv-ery systems and treatments for specific conditions of the human body.
Lines 183-186 - sentence with missing ending?
Answer: The sentence was corrected. Based on previous studies, rGO reduced using oxalic acid (rGO/AO) as red agent, has a rough surface, folded edges, ripples, multilayer structure, tangles of graphene layers and presence of ultra-thin and transparent graphene sheets as profile characteristic morphological. Furthermore, in view of this, it is noteworthy that the folds in the edge regions confer high tenacity to the material, therefore leading to excellent mechanical properties.
Line 447. betatin instead betanin
Answer: The error has been corrected.
Line 457 prGOressive
Answer: The error has been corrected.
Lines 461 - 467. Figure 9 made a part of some citation, or? it is neither clearly written nor explained.
Answer: The sequence of figures has been adjusted, with correct citation throughout the text.

Round 3
Reviewer 3 Report
Comments and Suggestions for Authors
Although the authors accept most of my suggestions, there are still a few minor things I would like to clarify.
1. You have misunderstood my suggestion Table 3, 4 and 5 - Heading: Equipaments ... Equipaments is wrong, equipment is correct.
Line 400 - is OG an error?
Line 402 - please leave out sucrose. Sucrose is not a reducing sugar. Leave only glucose in this sentence.
Line 413 - correct is 3,4-dihydroxycinnamic acid, please correct this /The other option is just hydroxycinnamic acid (without numbers, as a common name), but this is not so correct
Line 452 - more common is straight-chain instead of open-chain.
Author Response
Dear Editor,
We would like to express our sincere appreciation for your swift attention to our manuscript, IJMS-2952302, titled "Graphene and natural products: a review of antioxidant prop-erties in graphene oxide reduction". It is truly gratifying to witness a journal that prioritizes efficiency and expedience in its review process.
The suggestions presented in this new submission were fully met.
With my very best regards,
Filipe.
Reviewer 3
You have misunderstood my suggestion Table 3, 4 and 5 - Heading: Equipaments ... Equipaments is wrong, equipment is correct.
Answer: The error has been corrected.
Line 400 - is OG an error?
Answer: The error has been corrected.
Line 402 - please leave out sucrose. Sucrose is not a reducing sugar. Leave only glucose in this sentence.
Answer: The error has been corrected.
Line 413 - correct is 3,4-dihydroxycinnamic acid, please correct this /The other option is just hydroxycinnamic acid (without numbers, as a common name), but this is not so correct.
Answer: The error has been corrected.
Line 452 - more common is straight-chain instead of open-chain
Answer: The error has been corrected.
